# The Effect of Zinc, Selenium, and Their Combined Supplementation on Androgen Receptor Protein Expression in the Prostate Lobes and Serum Steroid Hormone Concentrations of Wistar Rats

**DOI:** 10.3390/nu12010153

**Published:** 2020-01-06

**Authors:** Adam Daragó, Michał Klimczak, Joanna Stragierowicz, Olga Stasikowska-Kanicka, Anna Kilanowicz

**Affiliations:** 1Department of Toxicology, Medical University of Lodz, Muszyńskiego 1, 90-151 Lodz, Poland; michal.klimczak@umed.lodz.pl (M.K.); joanna.stragierowicz@umed.lodz.pl (J.S.); 2Department of Nephropathology, Medical University of Lodz, Pomorska 251, 92-213 Lodz, Poland; olga.stasikowska@umed.lodz.pl

**Keywords:** rat, prostate, zinc, selenium, supplementation, androgen receptor, hormones

## Abstract

Background: Zinc (Zn) and selenium (Se) play a well-documented role in cancer prevention (e.g., for prostate cancer), and their combined supplementation is often given as a recommended prophylactic agent. The aim of the study was to determine the influence of Zn and/or Se supplementation on the androgen receptor (AR) in the prostate lobes and the serum selected hormone concentrations; a hitherto unresearched topic. Methods: Male rats (*n* = 84) were administered with Zn and/or Se intragastrically for up to 90 days. The effects of administration on the tested parameters were checked after 30 and 90 days of administration and additionally, 90 days after the end of 90 day administration. Results: Zn alone leads to an increase in serum testosterone concentrations, while the protein expression of AR in both parts of the prostate increases. Combined administration of Zn and Se eliminates the effect of Zn, which may suggest that these two elements act antagonistically. Se supplementation alone results in the same level of AR protein expression in administration and 90 days after administration periods. Conclusion: This paper presents the first report of the influence of Zn and/or Se supplementation on the protein expression of AR in the prostate. Our findings seem to indicate that simultaneous supplementation of both elements may be ineffective.

## 1. Introduction

Recent years have seen a worldwide increase in various forms of cancer, including prostate cancer (PCa). Both zinc (Zn) and selenium (Se) are well documented to counter oxidative stress, thus protecting DNA from the activity of reactive oxygen species [1]. It is now known that oxidative stress occurs in most pathological states and diseases [2], and oxidative imbalance in cells may lead to the oxidation of lipids, proteins, and DNA, which predispose to cancer transformation [3]. Therefore, supplementation with the two elements is often recommended as a prophylactic agent for various cancers, for example, for PCa, especially in men over 60 years of age [1,4]. 

Prostate gland is an organ regulated by androgens, mainly testosterone (T) and dihydrotestosterone (DHT). The role of androgens is essential to understand when considering the reasons and treatment of PCa [5,6], the second leading cause of death from cancer in the world [7,8]. Signaling of the androgen/androgen receptor (AR) is considered the driving force behind PCa [9]. Given its key role in the growth, proliferation, and survival of the prostate cells, it is not surprising that the androgen/AR signaling axis is crucial for prostate carcinogenicity and subsequent phases of disease progression [10]. In fact, androgen deprivation therapy has been the basis of PCa treatment for more than seven decades, since it was demonstrated that orchiectomy or high-dose estrogen treatment induced metastatic regression of PCa [11].

Studies on both animals and prostate cell lines have shown that Zn is an inhibitor of two key enzymes involved in T metabolism: 5α-reductase, which converts T to DHT, and aromatase, which is responsible for the formation of estradiol (E2) [12,13]. Therefore, as Zn appears to act as a modulator of the balance between T and DHT, it is typically included in supplementation strategies in patients with mild prostate hypertrophy [14]. Normal cells of peripheral zone accumulate high contents of Zn, in contrast to neoplastic, in which Zn concentration is remarkably lower, what results in reduced pro-apoptotic effects of this element. Therefore, it is hypothesized that increasing Zn concentrations can promote the apoptosis of malignant cells and inhibit neoplastic cell proliferation, migration, and invasion [15,16]. Similarly to Zn, Se is also used in anticancer supplementation, as it offers protection against free radicals, modifies immunity, DNA repair, apoptosis, and angiogenesis [17,18]. It is thought that Se prevents carcinogenesis by protecting the genome against oxidative damage and enhancing its repair [17]. Moreover, very importantly, Se acts as an essential element of T synthesis in male gonads [19]. 

Unfortunately, there is little data about the simultaneous influence of both elements on the functioning of the endocrinal system, especially in men. In addition, few studies in the literature have examined the interaction of Zn and Se [1,20]. However, cellular and molecular studies indicate that these two elements may interact with each other. Thus, controlling the balance between Zn and Se might be extremely important [17]. As was recently reviewed by Yildiz et al. (2019), Se can interrupt Zn homeostasis, which may result in disruption of metallothioneins (MTs) system, leading to oxidative damage of DNA or cancers [17,21]. Our previous research showed that the availability of Zn for the prostate gland varies depending on the supplementation variant used, that is, co-administration of Zn and Se affected the bioavailability of Zn in the prostate, which may indicate that Zn and Se should not be administered simultaneously [22]. 

Hence, the present in vivo animal study was performed probably for the first time to more precisely determine the effect of long-term supplementation with Zn and Se, either separately or combined, on the protein expression of AR in the prostate gland: AR is known to play a key role in maintaining the proper functioning of the prostate [23]. The study also examines the effect of supplementation on the serum concentrations of selected hormones: T, DHT, E2, luteinizing hormone (LH), and follicle stimulating hormone (FSH).

## 2. Material and Methods

### 2.1. Chemicals

Zinc gluconate (puriss grade) was purchased from Alfa Aesar GmbH & Co KG (Karlsruhe, Germany). Selenomethionine (puriss grade) was obtained from Sigma-Aldrich CHEMIE GmbH (Steinheim, Germany).

### 2.2. Animals

The study was carried out on 84 Wistar rats (six weeks old, weighing 290–320 g) from the breeding colony of the Medical University of Lodz, subjected to a two-week acclimatization period. The rats were housed under controlled temperature (22 ± 1 °C), relative humidity (45–55%), and a constant 12 h light/dark cycle. The animals were fed “Murigran”—a standard pelletized diet (Agropol, Poland). Food and water were supplied ad libitum throughout the study. The animals were divided randomly into three research groups receiving the following:
-zinc gluconate in conversion to Zn: 5 mg Zn/kg body weight (b.w.)/day (*n* = 21);-selenomethionine in conversion to Se: 2.8 µg Se/kg b.w./day (*n* = 21);-zinc gluconate and selenomethionine combined at the above doses (*n* = 21).

The applied doses were the same as in our previous study concerning Zn and Se bioavailability in the rat prostate [22]. In the case of Se, 2.8 µg/kg b.w. (0.7 µg/day/rat) corresponded to the average level of recommended dose used for supplementation in humans [24]. In the case of Zn gluconate, which is thought to be a nontoxic compound (median lethal dose LD_50_ ≥ 5000 mg/kg b.w., rat, oral route [25]), the dose corresponded to 1/1000 of LD_50_ (1/100 LD_50_ in conversion to Zn).

Zn gluconate and selenomethionine were dissolved in water and administered directly via intragastric gavage (per os) in the maximal volume of 0.5 mL per rat. Each experimental group comprised a control group (*n* = 21) consisting of animals receiving food and water ad libitum. Throughout both administration and postadministration periods, the behavior and look of all animals were observed.

After 30, 90, and 180 days (90 days administration period and 90 days observation period) from the start of the administration of the compounds, seven animals from each group were sacrificed, respectively, by intracardiac puncture under light carbon dioxide narcosis. Samples of whole blood (into Sarstedt tubes for metal analysis) and prostate gland, divided into ventral (V) and dorso-lateral (DL) parts, were collected. The blood was centrifuged, and the serum was frozen for further analysis. All tissues samples were fixed in 4% buffered formalin, and paraffin blocks were prepared.

The experiments were performed with the permission of the Local Ethical Committee for Experimentation on Animals (Resolutions No 43/LB 479/2009).

### 2.3. Immunohistochemical Staining and Scoring of AR 

Immunohistochemical staining was carried out on formalin-fixed paraffin-embedded rat sections according to a standard method. Briefly, 4 µm tissue sections were deparaffinized in xylene and rehydrated through a graded alcohol series. The samples were heated in a microwave oven in target retrieval solution pH 9.0 (TRS, Dako) for 30 min to retrieve antigens. Endogenous peroxidase was quenched with 3% hydrogen peroxide for 10 min. The sections were washed with TBS and incubated for 18 h with rabbit polyclonal primary antibody against androgen receptor (Abcam, UK; dilution 1:500; Cat No ab3510). After washing, EnVision-HRP detection system (Dako, Carpinteria, CA, USA) was used. 3,3′-diaminobenzidine was used as the chromogen. After counterstaining with Mayer’s hematoxylin, the slides were washed, dehydrated, cleared in xylene, and coverslipped. The primary antibody was replaced with antibody diluent for negative controls.

All sections were independently examined and scored by two pathomorphologists, who were blinded to the clinical information. The protein expression of AR (defined as a total score, TS) was scored by summing the mean signal intensity and the percentage of positively stained cells using Allred scale [26]. The AR immunoreactivity was categorized as positive when TS was ≥3. 

### 2.4. Hormone Determination

The determination of LH, FSH, and E2 in serum were performed with an Immulite 2000 automatic chemiluminescent immunoenzymatic analyzer (Siemens) using LH-PIL2KLH-19, FSH-PIL2KFS-13, and ESTRADIOL-PIL2KE2-24, respectively. All IMMULITE kits (Diagnostic Products Corporation, Los Angeles, CA, USA) are based on a solid-phase competitive immunoassay labeled with a chemiluminescent enzyme.

Serum T quantification was performed with the Elecsys Testosterone II test using the ECLIA electrochemiluminescence method on a Roche Elecsys 2010 Analyzer, while the quantity of DHT in serum was determined using an ELISA kit (Shanghai Sunred Biological Technology Co. Ltd., Shanghai, China; Product Code: 201-11-0564) on a Synergy ABS/HTR multisectional ELISA reader (Biotek, Winooski, VT, USA). Measurement ranges and sensitivity for the determination of the above hormones in rat serum are shown in Table 1.

### 2.5. Zinc and Selenium Determinations in Blood

Zn was determined by flame atomic absorption spectrometry (Avanta PM, GBC Scientific Equipment) after mineralization. A calibration curve was obtained using a range of concentrations of a standard solution of metals (ASTASOL, Prague, Czech Republic). The graph obtained was linear in the concentration range of 0.05–1.5 μg/mL, and the equation of the curve was as follows: *y* = 0.1967*x* + 0.0081; *R^2^* = 0.9969

The limit of detection, calculated as concentrations corresponding with an absorption value equal to a three-fold standard deviation of the signal for the lowest concentration, was 0.031 μg/mL. 

Se determinations was performed on a Hitachi F-4500 spectrofluorometer according to the method described by Danch and Drozdz (1996) [27]. The limit of detection was 0.010 μg/mL. The graph obtained was linear in the concentration range of 0.05–1.5 μg/mL, and the equation of the curve was as follows:*y* = 221.12*x* + 7.589; *R^2^* = 0.9980

The intra-laboratory quality control was based on certified standard lyophilized blood—Seronorm Whole blood L−1 (Sero AS, Billingstad, Norway). The reference material contained Zn and Se at concentrations of 4.6 μg/mL and 0.069 μg/mL (95% confidence interval: 3.8–5.3 μg/mL and 0.054–0.084 μg/mL), respectively. The relative standard deviations obtained in the reference material determinations were 2.3% for Zn and 5.5% for Se.

### 2.6. Statistical Analysis

Statistica 10.0 (StatSoft Inc., USA) was used for all statistical analyses. Bartlett’s test was used to confirm homogeneity of variance, following which Tukey’s test was used to determine the significance of the differences for the selected parameters. The statistical analysis of the immunoreactivity of AR in prostate lobes was limited to the TS and was performed based on two-factor analysis of variance. If the assumption about the homogeneity of variance was not fulfilled (no transformations stabilizing variance were applied), the resistance of F-Snedecor’s statistics was used, because the experimental system was completely balanced (in every time point and in every group there are seven observations). During the analysis, simple effects, that is, differences in rows and columns of the presented tables were evaluated. Differences with a *p*-value of less than 0.05 were considered statistically significant.

## 3. Results

During the whole experiment (administration and observation periods), no mortality and changes in animals behavior, look, and feed and water intake (data not shown) were noted. Figure 1 shows the percentage of rats with an AR-positive immunoreactivity in the V (Figure 1A) and DL (Figure 1B) prostate lobes after different administration periods and observation periods, while Table 2 shows the semiquantitative data of the protein expression of AR in the V and DL lobes of the rat prostate. Below 90 days exposure, nearly 60% of rats from the control group demonstrated positive AR stages in the DL lobe: three times higher than in the V part. However, after an additional 90 days observation, this value increased to 70% in V and 100% in DL.

Figure 2 shows representative examples of AR protein expression in control group (Figure 2A) after 30 days of Zn supplementation (Figure 2B,C), 90 days of Zn supplementation (Figure 2D,E), and 90 days of Se supplementation with an additional 90 day post-administration period (Figure 2F).

As shown in Table 2, Figure 1 and Figure 2D,E, the greatest statistically significant increase in AR protein expression in both parts of the prostate gland was observed after 90 day Zn supplementation. However, this effect was stronger in the V part of the prostate, and intensified with the duration of supplementation: being more than three times higher than control values after 30 days, and four times after 90 days. In the DL part, a statistically significant increase was observed only after 90 days. After the 90 day postadministration period, the AR protein expression levels in the Zn supplementation group had returned to control values in both parts of the prostate.

AR immunoreactivity did not differ significantly from control values at any measurement point, that is, after 30 or 90 days of Zn and Se supplementation, nor after the 90 day postadministration period (Table 2 and Figure 1). Moreover, in the group of animals receiving Se alone, the results also did not differ significantly from controls in either part of the prostate gland during the administration period. Interestingly, after the postadministration period, protein expression of AR in both lobes of rats administered with Se was significantly lower than that of the control rats, where an increase was observed (Figure 2F).

Table 3 presents the concentrations of selected hormones in serum: LH, FSH, E2, T, and DHT and also DHT/T and E/T ratio. Zn supplementation lasting up to 90 days, administered separately or together with Se, does not appear to significantly affect the secretion of the tested pituitary hormones. The concentrations of LH and FSH determined in serum during and after supplementation did not differ from control values.

As shown in Table 3, supplementation with Zn alone, without Se, resulted in an increase in T serum concentration during supplementation. Significantly higher T concentrations were observed: over 70% after the first 30 days of Zn administration and over 100% after 90 days. Such an increase in T concentrations during the supplementation period correlates with the observed increase in prostate AR immunoreactivity (Table 2 and Figure 1).

The calculated DHT/T ratio (Table 3) indirectly reflects the efficacy of Zn in transforming T into DHT. The ratio was significantly reduced, by about 30% compared with controls, after the first 30 days of Zn supplementation, and by about 60% after 90 days. Similarly, the ratio of E/T was significantly decreased, but only after 90 days of Zn exposure; this value persisted for the next 90 days after cessation of exposure.

Appendix A summarizes the results of Zn and Se concentrations in blood in all experimental groups of rats. The blood Zn concentrations did not significantly differ in any rats throughout the whole experiment, compared with controls. Statistically significant increase in Se concentration was noted only in the group of rats receiving Se alone for 90 days, compared with control group.

## 4. Discussion

One of the main factors regulating the androgen action in the prostate is AR. Its strongest ligand is DHT, and after binding, the two form an active DHT–AR complex. This complex penetrates into the cell nucleus, where it plays a crucial role both in the normal growth/development of the prostate gland, and in the progression of PCa [9,28]. Therefore, both benign prostate hyperplasia and PCa can be treated pharmacologically by inhibiting the conversion of T to DHT, or by blocking the action of AR [29]. 

Prostate cell proliferation is regulated by inter alia T and DHT on the one hand, and by Zrt/Irt-related protein 1 (ZIP1) transporter/Zn/citrate regulation on the other [13]. It has been proposed that Zn homeostasis disorders play key roles in the development of PCa, and that decreasing Zn levels may be one of the factors protecting cancer cells against apoptosis [13,16,30]. This may mean that maintaining the high Zn levels in the epithelial cells of peripheral zone, where prostate tumors are most frequently located, seems to be particularly important in PCa prophylaxis. Zn is also believed to maintain the homeostasis between T and DHT, and hence has been used for many years as a prophylactic supplement, together with Se, in men over 60 years old [14]. However, no studies have so far exhaustively examined the influence of both Zn and Se on the proper functioning of the prostate. While single studies have evaluated their individual effects on hormone levels and prostate cell proliferation or AR expression in the prostate, they have not been studied in combination [31]. Hence, it cannot be excluded that concurrent administration of Zn and Se may be not beneficial.

Our present study based on rats subjected to Zn and Se supplementation, both separately and in combination, found a significant increase in AR-positive immunoreactivity animals, but only in the groups receiving Zn alone. However, there is very little data on how Zn influences the expression/concentration of AR. Zn-deficient rats display decreased concentrations of AR in prostatic and hepatic cells, and decreased levels of DHT, T and LH in serum [32,33]. Although practically no data exists on AR expression following Zn administration, the opposite effect (an increase of AR protein expression) could be expected. Our present findings indicate that supplementation of rats with Zn significantly increases AR protein expression in both lobes of the prostate, but also that this effect was noted in the DL lobe only after 90 days of administration. It has previously been found that administration of Zn gluconate results in the accumulation of Zn in prostate tissue [22], which then inhibits the conversion of T to DHT by the inhibition of 5α-reductase. Similarly, the rats in the present study demonstrated increased serum T concentration, but not DHT concentration, which may have been associated with the presence of high Zn levels and greater AR protein expression. The calculated ratio of DHT to T may therefore provide an indirect measure of the effect of Zn as a modulator of T metabolism. In the present study, the ratio was reduced by about 30% in rats receiving Zn gluconate after the first 30 days of supplementation, and by about 60% after the next 60 days of supplementation. 

However, the actions of Zn on AR protein expression seem to be not so obvious. Recently, an in vitro study found Zn to suppress proliferation in AR-retaining PCa cells, but not in AR-deficient PCa cells. Zn not only suppressed AR expression in androgen-deprived conditions, but also down-regulated androgen-stimulated AR expression. On the other hand, in vivo studies provided more ambiguous results. In mice with implanted tumors (with TRAMP-C2 cells) after administration of 10 mg/kg ZnCl_2_, decreased AR protein expression was revealed. Interestingly, administration of this dose to mice without tumors did not result in such an effect. However, decreased AR protein expression was observed in mice without tumors after doubled dose of Zn, that is, 20 mg/kg, which turned out to be lethal to several mice with tumors [34]. The above results differ from our present findings, but it is difficult to explain opposite effects obtained in our study. However it should be underlined that we conducted an entirely distinct research model based on long-term (up to 90 days) supplementation (per os), so everyday administration of Zn may induce different effects in comparison with two single doses. Moreover, dose could be also another crucial factor. In our research, we applied a low dose (1/1000 LD_50_) of nontoxic compound (Zn gluconate, LD_50_ ≥ 5000 mg/kg b.w., rat, per os). Conversely, To et al. used more toxic compound (ZnCl_2_, LD_50_ = 24 mg/kg b.w., mice, intraperitoneal) in doses toxic to animals—half and almost LD_50_ [35]. As there are no other works on this issue, it is impossible to provide a more accurate explanation concerning the differences of AR protein expression. Moreover, the exact mechanism of Zn on PCa is still not well understood. Also, epidemiologic studies on Zn actions and risk of PCa are ambiguous and provide conflicting results. Recently, Gutierrez-Gonzalez et al. (2019) [15] proved that high dietary zinc intake could be associated with a higher PCa risk (especially for low-grade and localized tumors) [15].

In the case of Se, it was shown that this element is able to significantly reduce the transcription of AR and expression of prostate-specific agent (PSA). This down-regulation of AR signaling by Se serves as an important basis for the chemoprevention of PCa; however, its molecular mechanism remains unclear. The activation of AR requires binding to its ligand, translocation to the nucleus, and interaction with coregulators, including coactivators and corepressors, in the AR target genes [36]. Chun et al. (2006) suggest that Se may interfere with AR signaling at many stages, including AR mRNA expression, mRNA stability, protein degradation, nuclear translocation, and recruitment of coregulators [37]. The anticancer effect of Se has been confirmed by a decrease in PC3 tumor growth in mice [38] and by various in vitro studies in which Se inhibited the growth of prostate cancer cell lines, including LNCaP cells, which are sensitive to androgens, and DU145 and PC3 cells, which are insensitive [36,39,40].

Se has been found to reduce AR expression in rats supplemented with selenocysteine and isoflavones [41]. In the present study, the percentage of AR-immunopositive animals after Se supplementation was only about 20–30%. Moreover, these values did not differ from those observed in the control group within the administration period. Only 90 days after supplementation, that is, at the end of the postadministration period, protein expression of AR in both parts of the prostate was significantly lower than control values. Our in vivo results are consistent with the results of in vitro studies conducted by Dong et al. (2005) and Husbeck et al. (2006), which showed inhibition of AR expression and signaling by Se in human PCa cell lines [42,43].

## 5. Conclusions

Our findings confirm those of our previous work, indicating that simultaneous administration of Zn with Se may be not beneficial. Se affects the bioavailability of Zn to the prostate [22] and, as indicated in the present paper, it seems to suppress the effects of Zn on steroid hormone levels and AR protein expression in the prostate gland of rats. Therefore, the following conclusions can be drawn: firstly, Zn administration alone leads to an increase in T concentrations accompanied by an increase in the protein expression of AR in both parts of the prostate; secondly, combined administration of Zn and Se eliminates the effect of Zn on the protein expression of AR in the prostate and the serum concentration of steroid hormones in supplemented animals, which may suggest that the two elements act antagonistically; finally, Se supplementation alone results in the same level of AR protein expression in administration and 90 days after administration periods.

## Figures and Tables

**Figure 1 nutrients-12-00153-f001:**
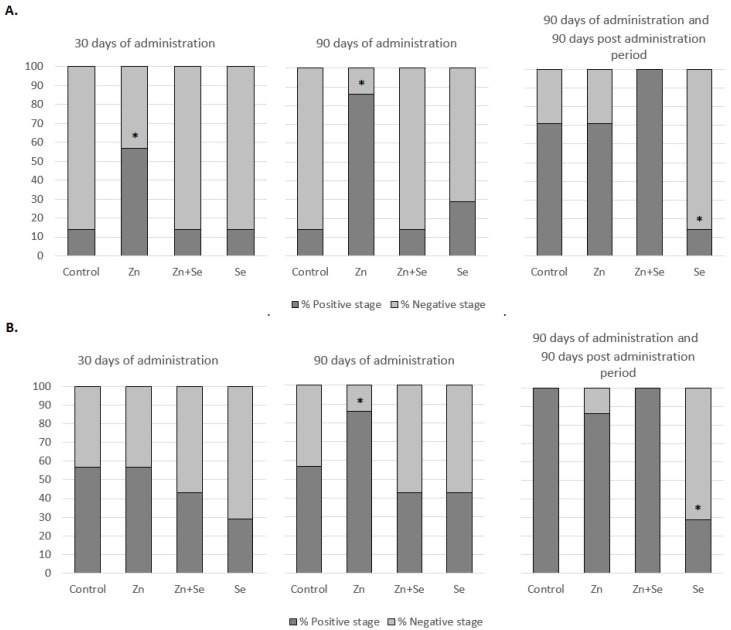
Percentage of rats with an androgen receptor (AR)-positive state in rat ventral (V) (**A**) and dorso-lateral (DL) (**B**) prostate lobe after 30 and 90 day administration of Zn and/or Se, and after 90 days exposure with an additional 90 day post-administration period. * results statistically significant in comparison with control group, *p* ≤ 0.05

**Figure 2 nutrients-12-00153-f002:**
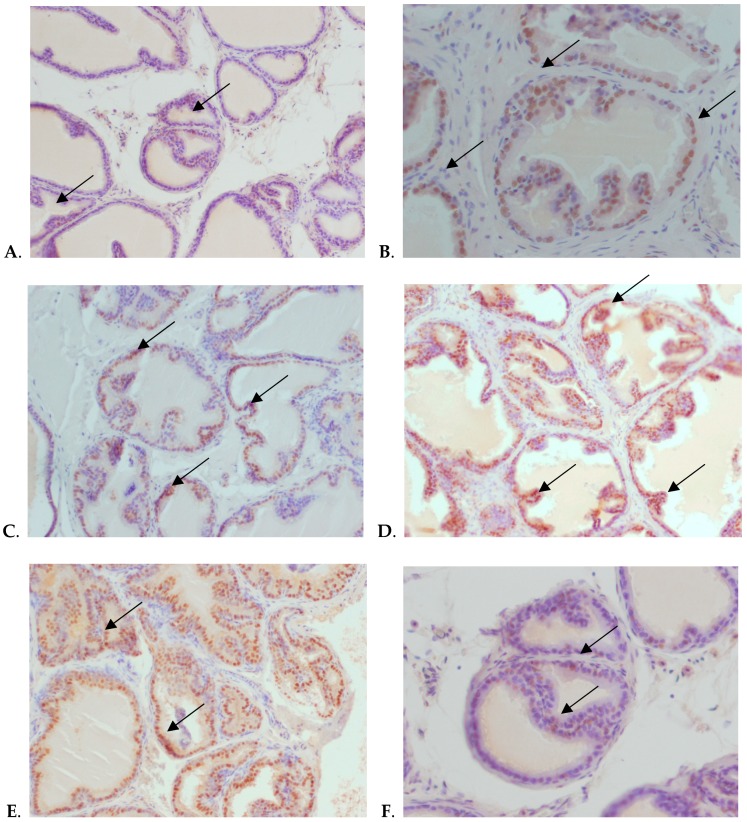
Immunohistochemical, nuclear localization of AR in the epithelial cells of V and DL prostatic lobes of the Wistar rats. (**A**) V, control, 30 days exposure, magnification ×200; (**B**) DL, Zn, 30 days exposure, magnification × 400; (**C**) V, Zn, 30 days exposure, magnification ×200; (**D**) V, Zn, 90 days exposure, magnification × 200; (**E**) DL, Zn, 90 days exposure, magnification ×200; (**F**) V, Se, 90 days exposure plus 90 days postexposure period, magnification ×400. The epithelial cells with brown nuclei are positive for AR (arrows), whereas epithelial cells with blue nuclei are negative for AR.

**Table 1 nutrients-12-00153-t001:** Analytical ranges and sensitivities of methods used to determine hormones.

	Analytical Range	Sensitivity
LH	0.052–10.4 ng/mL	2.6 pg/mL
FSH	0.052–8.9 ng/mL	5.2 pg/mL
E2	20–200 pg/mL	15 pg/mL
T	0.0125–15.0 ng/mL	0.12 ng/mL
DHT	0.3–72 pg/mL	0.224 pg/mL

LH: luteinizing hormone; FSH: follicle stimulating hormone; E2: estradiol; T: testosterone; DHT: dihydrotestosterone.

**Table 2 nutrients-12-00153-t002:** Means and medians of TS (total score) values for the rat prostate lobes (A, ventral; B, dorsolateral) after 30 and 90 day administration of Zn and/or Se, and after 90 days exposure with an additional 90 day post-administration period.

**A.**				
**Group**	**30 Days of Administration Mean ± SEM (Median)**	**90 Days of Administration Mean ± SEM (Median)**	**90 Days of Administration and 90-Day Post Administration Period Mean ± SEM (Median)**	**Comparing Means between Time Points in the Study Groups (*p*-Value)**
Control	1.57 ± 0.43 (2)	1.57 ± 0.43 (2)	2.86 ± 0.26 (3)	0.490
Zn	2.14 ± 0.59 (3)	3.14 ± 0.55 (4)	4.43 ± 0.78 (5)	0.201
Zn + Se	0.71 ± 0.47 (0)	0.71 ± 0.47 (0)	5.14 ± 0.59 (5)	0.002
Se	0.57 ± 0.57 (0)	2.14 ± 0.67 (2)	1.28 ± 0.47 (2)	0.336
Comparing groups (mean) at the same time points (*p*-value)	0.105	0.026	0.015	Interaction test: *p* = 0.217
**B.**				
**Group**	**30 Days of Administration Mean ± SEM (Median)**	**90 Days of Administration Mean ± SEM (Median)**	**90 Days of Administration and 90 Day Post Administration Period Mean ± SEM (Median)**	**Comparing Means between Time Points in the Study Groups (*p*-Value)**
Control	2.29 ± 0.64 (3)	2.86 ± 0.59 (4)	4.86 ± 0.26 (5)	0.015
Zn	2.57 ± 0.78 (3)	4.14 ± 0.59 (4)	4.86 ± 0.98 (5)	0.023
Zn + Se	1.86 ± 0.70 (2)	2.57 ± 0.30 (2)	6.14 ± 0.26 (6)	<0.005
Se	1.86 ± 0.99 (0)	2.57 ± 0.75 (3)	1.57 ± 1.02 (0)	0.365
Comparing groups (mean) at the same time points (*p*-value)	0.545	0.036	<0.005	Interaction test: *p* = 0.061

**Table 3 nutrients-12-00153-t003:** Serum LH, FSH, E2, T, and DHT concentrations and the ratio of DHT/T and E/DHT (mean ± SD) after 30 and 90 day administration of Zn and/or Se, and after 90 days exposure with an additional 90 day postadministration period.

		LH [ng/mL]	FSH [ng/mL]	E2 [pg/mL]	T [ng/mL]	DHT [pg/mL]	DHT/T [* 10^3^]	E/T [* 10^3^]
30 days of administration	Control	0.68 ± 0.28	2.2 ± 0.55	25 ± 5.4	3.6 ± 0.75	5.50 ± 1.3	1.4 ± 0.19	7.0 ± 2.3
	Zn	0.54 ± 0.41	2.0 ± 0.43	28 ± 7.8	4.8 ± 0.49 *	4.5 ± 0.70	1.0 ± 0.11 *	5.8 ± 2.5
	Se	0.64 ± 0.28	2.5 ± 0.49	34 ± 5.3	4.5 ± 1.5	5.3 ± 1.7	1.1 ± 0.34	7.5 ± 2.2
	Zn + Se	0.73 ± 0.14	2.7 ± 0.54	31 ± 5.4	3.1 ± 1.6	4.6 ± 0.80	1.5 ± 0.24	10.2 ± 3.1
90 days of administration	Control	0.73 ± 0.21	2.3 ± 0.48	37 ± 6.6	5.7 ± 0.81	5.4 ± 0.75	1.0 ± 0.14	6.8 ± 2.1
	Zn	0.35 ± 0.43	2.9 ± 0.55	33 ± 7.1	13 ± 6.4 *	4.8 ± 0.89	0.4 ± 0.4 *	2.4 ± 2.2 *
	Se	0.81 ± 0.31	2.0 ± 0.15	30 ± 11	6.4 ± 1.2	5.4 ± 1.0	0.8 ± 0.21	4.7 ± 2.8
	Zn + Se	0.82 ± 0.32	2.8 ± 0.33	30 ± 6.8	3.1 ± 1.0	4.9 ± 1.6	1.4 ± 0.28	9.8 ± 2.8
90 days of administration and 90 day post administration period	Control	0.77 ± 0.21	2.2 ± 0.41	47 ± 9.1	2.2 ± 1.8	5.1 ± 0.69	2.3 ± 0.36	21.6 ± 3.7
	Zn	0.72 ± 0.15	2.1 ± 0.31	44 ± 18	3.7 ± 2.1	5.2 ± 0.75	1.6 ± 0.38	11.8 ± 4.8 *
	Se	0.68 ± 0.13	2.4 ± 0.31	38 ± 11	2.7 ± 1.6	4.5 ± 1.6	1.8 ± 0.31	14.0 ± 4.2
	Zn + Se	0.74 ± 0.19	2.2 ± 0.21	40 ± 5.4	2.6 ± 0.46	5.3 ± 1.8	2.1 ± 0.39	15.5 ± 3.2

* results statistically significant in comparison with control group, *p* ≤ 0.05.

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
