# Peer review of "The Effect of Zinc, Selenium, and Their Combined Supplementation on Androgen Receptor Protein Expression in the Prostate Lobes and Serum Steroid Hormone Concentrations of Wistar Rats"

_nutrients, 2020, doi:10.3390/nu12010153_

Round 1

Reviewer 1 Report

Authors are suggested to label the figure 2 and indicate arrows to display the zn and se effect before and after treatment on immune expression of AR

Authors are suggested to include details of statistical analysis in methods as well as in results

Author Response

Dear Reviewer 1,

Thank for your favourable review of our manuscript and valuable comments. We have tried to respect all suggestions given by you in the review. Below you will find the description of our amendments.

1) Authors are suggested to label the figure 2 and indicate arrows to display the zn and se effect before and after treatment on immune expression of AR

Answer: We revised and extended the description of Figure 2. We hope that the proposed change will make it easier to read the presented results.

2) Authors are suggested to include details of statistical analysis in methods as well as in results

Answer: The statistical analysis section was included in the previous version of the manuscript, but it was unfortunately mistakenly presented below the Table 1. In the revised version details of statistical analysis are put in the section “Statistical analysis”. Moreover we unified and revised statistical reference under figures and tables.

In the corrected version of the manuscript, we have also respected your other suggestions.

Reviewer 2 Report

1) The tile is not informative, which should be more clear statement;

2) The authors have not articulated why they conducted this study with novelty and originality

3) Some key information is missing such as the prevention mechanisms of Zn/Se for prostate cancer in vitro or in vivo

4) The concentrations of  Zn and Se in serum or hair in the rats with administrations should be monitored

5) Please clarify the concentrations of Zn and Se administrated to rats are from experience or literature, and whether the linear dosages needed

6) The explanations of the differences of the findings of this study and literature are not so convincing

7) The protein level of AR could be presented as "the expression/the protein level/the protein expression".  The immunoexpression is not commonly used in US and causes confusion

8) The Ar expression doesn't equal to Ar signaling, the evaluations of Ar responsive genes in prostate epithelial cells should be included

Author Response

Dear Reviewer 2,

Thank you very much for your careful review of our manuscript. We have analysed all the valuable comments thoroughly. Below you will find the description of our amendments.

1) The title is not informative, which should be more clear statement;

Answer: According to your suggestion we change the title into “The effect of zinc, selenium and their combined supplementation on androgen receptor protein expression in the prostate lobes and serum steroid hormone concentrations of Wistar rats”. We think that now it better reflects the aim of this study and now is more informative.

2) The authors have not articulated why they conducted this study with novelty and originality

Answer: We have changed the layout of the introduction and revised this section to highlight the elements of novelty in our research and we underlined, why we conduct this study. We hope that amends of this section will be satisfactory.

3) Some key information is missing such as the prevention mechanisms of Zn/Se for prostate cancer in vitro or in vivo

Answer: Thank you for this suggestion. We added basic mechanisms of Zn/Se prevention in prostate cancer in the introduction section.

4) The concentrations of Zn and Se in serum or hair in the rats with administrations should be monitored

Answer: The blood levels of Zn and Se were determined. Since these results were comparable to those presented in the previous paper (Daragó A., Sapota A., Nasiadek M., Klimczak M., Kilanowicz A. The Effect of Zinc and Selenium Supplementation Mode on Their Bioavailability in the Rat Prostate. Should Administration Be Joint or Separate? Nutrients, 2016, 8(10), E601), we have not provide them in the present paper. However, as you suggested, we added the data in the supplementary material and we described the results of this determinations in the paper. Moreover, methods of determination of these elements were also included in the Material and methods section.

5) Please clarify the concentrations of Zn and Se administrated to rats are from experience or literature, and whether the linear dosages needed

Answer: Thank you very much for this comment, because we found a mistake in presented dose. The applied doses are the same as in our previous work, so we amend the dose into Zn: 5 mg Zn/kg b.w./day and Se: 2.8 µg Se/kg b.w./day. As we write in the material and methods section: “In the case of Se, 2.8 µg/kg b.w. (0.7 µg/day/rat) corresponded to average level of recommended dose used for supplementation in humans. In the case of Zn gluconate, which is thought to be a nontoxic compound (LD50 ≥ 5000 mg/kg b.w., rat, oral route), the dose corresponded to 1/1000 of LD50 (1/100 LD50 in conversion to Zn)”.

6) The explanations of the differences of the findings of this study and literature are not so convincing

Answer: According to your suggestion we revised the discussion section. It is very difficult to explain these differences. There is very little data on this issue in the literature and, what is important, the role of Zn seems to be not so obvious. However, we have tried to discuss this issue more thoroughly to make our explanation more convincing.

7) The protein level of AR could be presented as "the expression/the protein level/the protein expression". The immunoexpression is not commonly used in US and causes confusion

Answer: According to your suggestion, we have changed "immunoexpression" to "protein expression" in the whole manuscript.

8) The Ar expression doesn't equal to Ar signaling, the evaluations of Ar responsive genes in prostate epithelial cells should be included.

Answer: We agree with your opinion that the AR expression is not equal to AR signalling. However, we have not performed evaluations of AR responsive genes in prostate epithelial cells and we will consider this in our future studies concerning the role of Zn and Se on AR signalling.

Round 2

Reviewer 2 Report

Thank you for providing the editing and explanation as requested.